# Optimal decision-making with time-varying evidence reliability

**Jan Drugowitsch**[1]      **Rubén Moreno-Bote**[2]      **Alexandre Pouget**[1]

[1]Dépt. des Neurosciences Fondamentales
Université de Genève
CH-1211 Genève 4, Switzerland
jdrugo@gmail.com,
alexandre.pouget@unige.ch

[2]Research Unit, Parc Sanitari
Sant Joan de Déu and
University of Barcelona
08950 Barcelona, Spain
rmoreno@fsjd.org

## Abstract

Previous theoretical and experimental work on optimal decision-making was restricted to the artificial setting of a reliability of the momentary sensory evidence that remained constant within single trials. The work presented here describes the computation and characterization of optimal decision-making in the more realistic case of an evidence reliability that varies across time even within a trial. It shows that, in this case, the optimal behavior is determined by a bound in the decision maker's belief that depends only on the current, but not the past, reliability. We furthermore demonstrate that simpler heuristics fail to match the optimal performance for certain characteristics of the process that determines the time-course of this reliability, causing a drop in reward rate by more than 50%.

## 1 Introduction

Optimal decision-making constitutes making optimal use of sensory information to maximize one's overall reward, given the current task contingencies. Example of decision-making are the decision to cross the road based on the percept of incoming traffic, or the decision of an eagle to dive for prey based on the uncertain information of the prey's presence and location. Any kind of decision-making based on sensory information requires some temporal accumulation of this information, which makes such accumulation the first integral component of decision-making. Accumulating evidence for a longer duration causes higher certainty about the stimulus but comes at the cost of spending more time to commit to a decision. Thus, the second integral component of such decision-making is to decide when enough information has been accumulated to commit to a decision.

Previous work has established that, if the reliability of momentary evidence is constant within a trial but might vary across trials, optimal decision-making can be implemented by a class of models known as *diffusion models* [1, 2, 3]. Furthermore, it has been shown that the behavior of humans and other animals at least qualitatively follow that predicted by such diffusion models [4, 5, 6, 3].

Our work significantly extends this work by moving from the rather artificial case of constant evidence reliability to allowing the reliability of evidence to change within single trials. Based on a principled formulation of this problem, we describe optimal decision-making with time-varying evidence reliability. Furthermore, a comparison to simpler decision-making heuristics demonstrates when such heuristics fail to feature comparable performance. In particular, we derive Bayes-optimal evidence accumulation for our task setup, and compute the optimal *policy* for such cases by dynamic programming. To do so, we borrow concepts from continuous-time stochastic control to keep the computational complexity linear in the process space size (rather than quadratic for the naïve approach). Finally, we characterize how the optimal policy depends on parameters that determine the evidence reliability time-course, and show that simpler, heuristic policies fail to match the optimal performance for particular sub-regions of this parameter space.

## 2 Perceptual decision-making with time-varying reliability

Within a single trial, the decision maker's task is to identify the state of a binary hidden variable, $z \in \{-1, 1\}$ (with units $s^{-1}$, if time is measured in seconds), based on a stream of momentary evidence $dx(t)$, $t \geq 0$. This momentary evidence provides uncertain information about $z$ by

$$dx = zdt + \frac{1}{\sqrt{\tau(t)}}dW, \quad \text{where } d\tau = \eta\left(\mu - \tau\right)dt + \sigma\sqrt{\frac{2\eta}{\mu}}\sqrt{\tau}dB, \qquad (1)$$

where $dW$ and $dB$ are independent Wiener processes. In the above, $\tau(t)$ controls how informative the *momentary evidence* $dx(t)$ is about $z$, such that $\tau(t)$ is the *reliability* of this momentary evidence. We assume its time-course to be described by the Cox-Ingersoll-Ross (CIR) process ($\tau(t)$ in Eq. (1)) [7]. Despite the simplicity of this model and its low number of parameters, it is sufficiently flexible in modeling how the evidence reliability changes with time, and ensures that $\tau \geq 0$, always[1]. It is parameterized by the mean reliability, $\mu$, its variance, $\sigma^2$, and its speed of change, $\eta$, all of which we assume to be known to the decision maker. At the beginning of each trial, at $t = 0$, $\tau(0)$ is drawn from the process' steady-state distribution, which is gamma with shape $\mu^2/\sigma^2$ and scale $\sigma^2/\mu$ [7]. It can be shown, that upon observing some momentary evidence, $\tau(t)$ can be immediately estimated with infinite precision, such that it is known for all $t \geq 0$ (see supplement).

Optimal decision-making requires in each trial computing the posterior $z$, given all evidence $dx_{0:t}$ from trial onset to some time $t$. Assuming a uniform prior over $z$'s, this posterior is given by

$$g(t) \equiv p\left(z = 1 | dx_{0:t}\right) = \frac{1}{1 + e^{-2X(t)}}, \qquad \text{where } X(t) = \int_0^t \tau(s)dx(s), \qquad (2)$$

(this has already been established in [8]; see supplement for derivation). Thus, at time $t$, the decision maker's *belief* $g(t)$ that $z = 1$ is the sigmoid of the accumulated, reliability-weighted, momentary evidence up until that time.

We consider two possible tasks. In the $ER$ task, the decision maker is faced with a single trial in which correct (incorrect) decisions are rewarded by $r^+$ ($r^-$), and the accumulation of evidence comes at a constant cost (for example, attentional effort) of $c$ per unit time. The decision maker's aim is then to maximize her expected reward, $ER$, including the cost for accumulating evidence. In the $RR$ task, we consider a long sequence of trials, separated on average by the inter-trial interval $t_i$, which might be extended by the penalty time $t_p$ for wrong decisions. Maximizing reward in such a sequence equals maximizing the reward rate, $RR$, per unit time [9]. Thus, the objective function for either task is given by

$$ER\left(PC, DT\right) = PCr^+ + (1 - PC)r^- - cDT, \quad RR\left(PC, DT\right) = \frac{ER\left(PC, DT\right)}{DT + t_i + (1 - PC)t_p}, \quad (3)$$

where $PC$ is the probability of performing a correct decision, and $DT$ is the expected decision time. For notational convenience we assume $r^+ = 1$ and $r^- = 0$. The work can be easily generalized to any choice of $r^+$ and $r^-$.

## 3 Finding the optimal policy by Dynamic Programming

### 3.1 Dynamic Programming formulation

Focusing first on the $ER$ task of maximizing the expected reward in a single trial, the optimal policy can be described by bounds in belief[2] at $g_\theta(\tau)$ and $1 - g_\theta(\tau)$ as functions of the current reliability, $\tau$. Once either of these bounds is crossed, the decision maker chooses $z = 1$ (for $g_\theta(\tau)$) or $z = -1$ (for $1 - g_\theta(\tau)$). The bounds are found by solving Bellman's equation [10, 9],

$$V(g, \tau) = \max\left\{V_d(g), \langle V(g + \delta g, \tau + \delta\tau)\rangle_{p(\delta g, \delta\tau | g, \tau)} - c\delta t\right\}, \qquad (4)$$

where $V_d(g) = \max\{g, 1 - g\}$. Here, the value function $V(g, \tau)$ denotes the *expected return* for current *state* $(g, \tau)$ (i.e. holding belief $g$, and current reliability $\tau$), which is the expected reward at

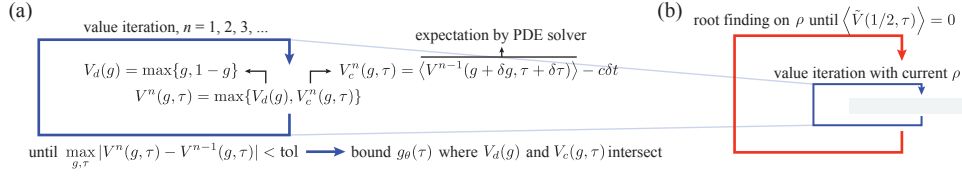

Figure 1: Finding the optimal policy by dynamic programming. (a) illustrates the approach for the $ER$ task. Here, $V_d(g)$ and $V_c(g, \tau)$ denote the expected return for immediate decisions and that for continuing to accumulate evidence, respectively. (b) shows the same approach for $RR$ tasks, in which, in an outer loop, the reward rate $\rho$ is found by root finding.

this state within a trial, given that optimal choices are performed in all future states. The right-hand side of Bellman's equation is the maximum of the expected returns for either making a decision immediately, or continuing to accumulate more evidence and deciding later. When deciding immediately, one expects reward $g$ (or $1 - g$) when choosing $z = 1$ (or $z = -1$), such that the expected return for this choice is $V_d(g)$. Continuing to accumulate evidence for another small time step $\delta t$ comes at cost $c\delta t$, but promises future expected return $\langle V(g + \delta g, \tau + \delta\tau) \rangle_{p(\delta g, \delta\tau | g, \tau)}$, as expressed by the second term in $\max\{\cdot, \cdot\}$ in Eq. (4). Given a $V(g, t)$ that satisfies Bellman's equation, it is easy to see that the optimal policy is to accumulate evidence until the expected return for doing so is exceeded by that for making immediate decisions. The belief $g$ at which this happens differs for different reliabilities $\tau$, such that the optimal policy is determined by a bound in belief, $g_\theta(\tau)$, that depends on the current reliability.

We find the solution to Bellman's equation itself by value iteration on a discretized $(g, \tau)$-space, as illustrated in Fig. 1(a). Value iteration is based on a sequence of value functions $V^0(g, \tau), V^1(g, \tau), \ldots$, where $V^n(g, \tau)$ is given by the solution to right-hand side of Eq. (4) with $\langle V(g + \delta g, \tau + \delta\tau) \rangle$ based on the previous value function $V^{n-1}(g, \tau)$. With $n \to \infty$, this procedure guarantees convergence to the solution of Eq. (4). In practice, we terminate value iteration once $\max_{g,\tau} |V^n(g, \tau) - V^{n-1}(g, \tau)|$ drops below a pre-defined threshold. The only remaining difficulty is how to compute the expected future return $\langle V(\cdot, \cdot) \rangle$ on the discretized $(g, \tau)$-space, which we describe in more detail in the next section.

The $RR$ task, in which the aim is to maximize the reward rate, requires the use of average-reward Dynamic Programming [9, 11], based on the *average-adjusted expected return*, $\tilde{V}(g, \tau)$. If $\rho$ denotes the reward rate (avg. reward per unit time, $RR$ in Eq. (3)), this expected return penalizes the passage of some time $\delta t$ by $-\rho\delta t$, and can be interpreted as how much better or worse the current state is than the average. It is relative to an arbitrary baseline, such that adding a constant to this return for all states does not change the resulting policy [11]. We remove this additional degree of freedom by fixing the average $\tilde{V}(\cdot, \cdot)$ at the beginning of a trial (where $g = 1/2$) to $\langle \tilde{V}(1/2, \tau) \rangle_{p(\tau)} = 0$, where the expectation is with respect to the steady-state distribution of $\tau$. Overall, this leads to Bellman's equation,

$$\tilde{V}(g, \tau) = \max\left\{ \tilde{V}_d(g), \left\langle \tilde{V}(g + \delta g, \tau + \delta\tau) \right\rangle_{p(\delta g, \delta\tau | g, \tau)} - (c + \rho)\delta t \right\} \tag{5}$$

with the average-adjusted expected return for immediate decisions given by

$$\tilde{V}_d(g) = \max\left\{ g - \rho\left(t_i + (1 - g)t_p\right), 1 - g - \rho\left(t_i + gt_p\right) \right\}. \tag{6}$$

The latter results from a decision being followed by the inter-trial interval $t_i$ and an eventual penalty time $t_p$ for incorrect choices, after which the average-adjusted expected return is $\langle \tilde{V}(1/2, \tau) \rangle = 0$, as previously chosen. The value function is again computed by value iteration, assuming a known $\rho$. The correct $\rho$ itself is found in an outer loop, by root-finding on the consistency condition, $\langle \tilde{V}(1/2, \tau) \rangle = 0$, as illustrated in Fig. 1(b).

### 3.2 Finding $\langle V(g + \delta g, \tau + \delta\tau) \rangle$ as solution to a PDE

Performing value iteration on Eq. (4) requires computing the expectation $\langle V(g + \delta g, \tau + \delta\tau) \rangle_{p(\delta g, \delta\tau | g, \tau)}$ on a discretized $(g, \tau)$ space. Naïvely, we could perform the

required integration by the rectangle method or related methods, but this has several disadvantages. First, the method scales quadratically in the size of the $(g, \tau)$ space. Second, with $\delta t \to 0$, $p(\delta g, \delta \tau | g, \tau)$ becomes singular, such that small time discretization requires even smaller state discretization. Third, it requires explicit computation of $p(\delta g, \delta \tau | g, \tau)$, which might be cumbersome.

Instead, we borrow methods from stochastic optimal control [12] to find the expectation as a solution to the partial differential equation (PDE). To do so, we link $V(g, \tau)$ to $\langle V(g + \delta g, \tau + \delta \tau) \rangle$, by considering how $g$ and $\tau$ evolve from some time $t$ to time $t + \delta t$. Defining $u(g, \tau, t) \equiv V(g, \tau)$ and $u(g, \tau, t + \delta t) \equiv \langle V(g + \delta g, \tau + \delta \tau) \rangle$, and replacing this expectation by its second-order Taylor expansion around $(g, \tau)$, we find that, with $\delta t \to 0$, we have

$$\frac{\partial u}{\partial t} = \left( \frac{\langle \mathrm{d}g \rangle}{\mathrm{d}t} \frac{\partial}{\partial g} + \frac{\langle \mathrm{d}\tau \rangle}{\mathrm{d}t} \frac{\partial}{\partial \tau} + \frac{\langle \mathrm{d}g^2 \rangle}{2\mathrm{d}t} \frac{\partial^2}{\partial g^2} + \frac{\langle \mathrm{d}\tau^2 \rangle}{2\mathrm{d}t} \frac{\partial^2}{\partial \tau^2} + \frac{\langle \mathrm{d}g \mathrm{d}\tau \rangle}{\mathrm{d}t} \frac{\partial^2}{\partial g \partial \tau} \right) u, \qquad (7)$$

with all expectations implicitly conditional on $g$ and $\tau$. If we approximate the partial derivatives with respect to $g$ and $\tau$ by their central finite differences, and denote $u_{kj}^n \equiv u(g_k, \tau_j, t)$ and $u_{kj}^{n+1} \equiv u(g_k, \tau_j, t + \delta t)$ ($g_k$ and $\tau_j$ are the discretized state nodes), applying the Crank-Nicolson method [13] to the above PDE results in the linear system

$$\boldsymbol{L}^{n+1} \boldsymbol{u}^{n+1} = \boldsymbol{L}^n \boldsymbol{u}^n \qquad (8)$$

where both $\boldsymbol{L}^n$ and $\boldsymbol{L}^{n+1}$ are sparse matrices, and the $\boldsymbol{u}$'s are vectors that contain all $u_{kj}$. Computing $\langle V(g + \delta g, \tau + \delta \tau) \rangle$ now conforms to solving the above linear system with respect to $\boldsymbol{u}^{n+1}$. As the process on $g$ and $\tau$ only appears as its infinitesimal moments in Eq. (7), this approach neither requires explicit computation of $p(\delta g, \delta \tau | g, \tau)$ nor suffers from singularities in this density. It still scales quadratically with the state space discretization, but we achieve linear scaling by switching from the Crank-Nicolson to the Alternating Direction Implicit (ADI) method [13] (see supplement for details). This method splits the computation into two steps of size $\delta t / 2$, in each of which the partial derivatives are only implicit with respect to one of the two state space dimensions. This results in a tri-diagonal structure of the linear system, and an associated reduction of the computational complexity while preserving the numerical robustness of the Crank-Nicholson method [13].

The PDE approach requires us to specify how $V$ (and thus $u$) behaves at the boundaries, $g \in \{0, 1\}$ and $\tau \in \{0, \infty\}$. Beliefs $g \in \{0, 1\}$ imply complete certainty about the latent variable $z$, such that a decision is imminent. This implies that, at these beliefs, we have $V(g, \tau) = V_d(g)$ for all $\tau$. With $\tau \to \infty$, the reliability of the momentary evidence becomes overwhelming, such that the latent variable $z$ is again immediately known, resulting in $V(g, \tau) \to V_d(1) (= V_d(0))$ for all $g$. For $\tau = 0$, the infinitesimal moments are $\langle \mathrm{d}g \rangle = \langle \mathrm{d}g^2 \rangle = \langle \mathrm{d}\tau^2 \rangle = 0$, and $\langle \mathrm{d}\tau \rangle = \eta \mu \mathrm{d}t$, such that $g$ remains unchanged and $\tau$ drifts deterministically towards positive values. Thus, there is no leakage of $V$ towards $\tau < 0$, which makes this lower boundary well-defined.

## 4 Results

We first provide an example of an optimal policy and how it shapes behavior, followed by how different parameters of the process on the evidence reliability $\tau$ and different task parameters influence the shape of the optimal bound $g_\theta(\tau)$. Then, we compare the performance of these bounds to the performance that can be achieved by simple heuristics, like the diffusion model with a constant bound, or a bound in belief independent of $\tau$.

In all cases, we computed the optimal bounds by dynamic programming on a $200 \times 200$ grid on $(g, \tau)$, using $\delta t = 0.005$. $g$ spun its whole $[0, 1]$ range, and $\tau$ ranged from 0 to twice the 99th percentile of its steady-state distribution. We used $\max_{g, \tau} |V^n(g, \tau) - V^{n-1}(g, \tau)| \leq 10^{-3} \delta t$ as convergence criterion for value iteration.

### 4.1 Decision-making with reliability-dependent bounds

Figure 2(a) shows one example of an optimal policy (black lines) for an $ER$ task with evidence accumulation cost of $c = 0.1$ and $\tau$-process parameters $\mu = 0.4$, $\sigma = 0.2$, and $\eta = 1$. This policy can be understood as follows. At the beginning of each trial, the decision maker starts at

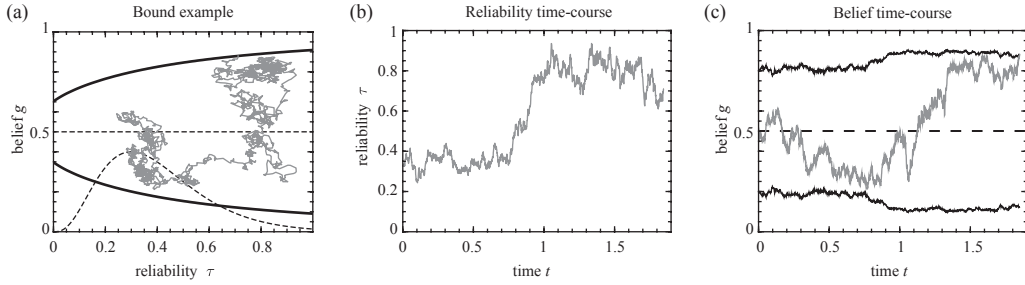

Figure 2: Decision-making with the optimal policy. (a) shows the optimal bounds, at $g_\theta(\tau)$ and $1 - g_\theta(\tau)$ (black) and an example trajectory (grey). The dashed curve shows the steady-state distribution of the $\tau$-process. (b) shows the $\tau$-component (evidence reliability) of this example trajectory over time. Even though not a jump-diffusion process, the CIR process can feature jump-like transitions — here at around 1s. (c) shows the $g$-component (belief) of this trajectory over time (grey), and how the change in evidence reliability changes the bounds on this belief (black). Note that the bound fluctuates rapidly due to the rapid fluctuation of $\tau$, even though the bound itself is continuous in $\tau$.

$g(0) = 1/2$ and some $\tau(0)$ drawn from the steady-state distribution over $\tau$'s (dashed curve in Fig. 2(a)). When accumulating evidence, the decision maker's belief $g(t)$ starts diffusing and drifting towards either 1 or 0, following the dynamics described in Eqs. (1) and (2). At the same time, the reliability $\tau(t)$ changes according to the CIR process, Eq. (1) (Fig. 2(b)). In combination, this leads to a two-dimensional trajectory in the $(g, \tau)$ space (Fig. 2(a), grey line). A decision is reached once this trajectory reaches either $g_\theta(\tau)$ or $1 - g_\theta(\tau)$ (Fig. 2(a), black lines). In belief space, this corresponds to a bound that changes with the current reliability. For the example trajectory in Fig. 2, this reliability jumps to higher values after around $1s$ (Fig. 2(b)), which leads to a corresponding jump of the bound to higher levels of confidence (black line in Fig. 2(c)).

In general, the optimal bound is an increasing function in $\tau$. Thus, the larger the current reliability of the momentary evidence, the more sense it makes to accumulate evidence to a higher level of confidence before committing to a choice. This is because a low evidence reliability implies that – at least in the close future – this reliability will remain low, such that it does not make sense to pay the cost for accumulating evidence without the associated gain in choice accuracy. A higher evidence reliability implies that high levels of confidence, and associated choice accuracy, are reached more quickly, and thus at a lower cost. This also indicates that a decision bound increasing in $\tau$ does *not* imply that high-reliability evidence will lead to slower choices. In fact, the opposite is true, as a faster move towards higher confidence for high reliability causes faster decisions in such cases.

## 4.2 Optimal bounds for different reliability/task parameters

To see how different parameters of the CIR process on the reliability influence the optimal decision bound, we compared bounds where one of its parameters is systematically varied. In all cases, we assumed an $ER$ task with $c = 0.1$, and default CIR process parameters $\mu = 0.4$, $\sigma = 0.2$, $\eta = 2$.

Figure 3(a) shows how the bound differs for different means $\mu$ of the CIR process. A lower mean implies that, on average, the task will be harder, such that more evidence needs to be accumulated to reach the same level of performance. This accumulation comes at a cost, such that the optimal policy is to stop accumulating earlier in harder tasks. This causes lower decision bounds for smaller $\mu$. Fig. 3(b) shows that the optimal bound only very weakly depends on the standard deviation $\sigma$ of the reliability process. This standard deviation determines how far $\tau$ can deviate from its mean, $\mu$. The weak dependence of the bound on this parameter shows that it is not that important to which degree $\tau$ fluctuates, as long as it fluctuates with the same speed, $\eta$. This speed has a strong influence on the optimal bound, as shown in Fig. 3(c). For a slowly changing $\tau$ (low $\eta$), the current $\tau$ is likely to remain the same in the future, such that the optimal bound strongly depends on $\tau$. For a rapidly changing $\tau$, in contrast, the current $\tau$ does not provide much information about future reliabilities, such that the optimal bound features only a very weak dependence on the current evidence reliability.

Similar observations can be made for changes in task parameters. Figure 3(d) illustrates that a larger cost $c$ generally causes lower bounds, as it pays less to accumulate evidence. In $RR$ tasks, the

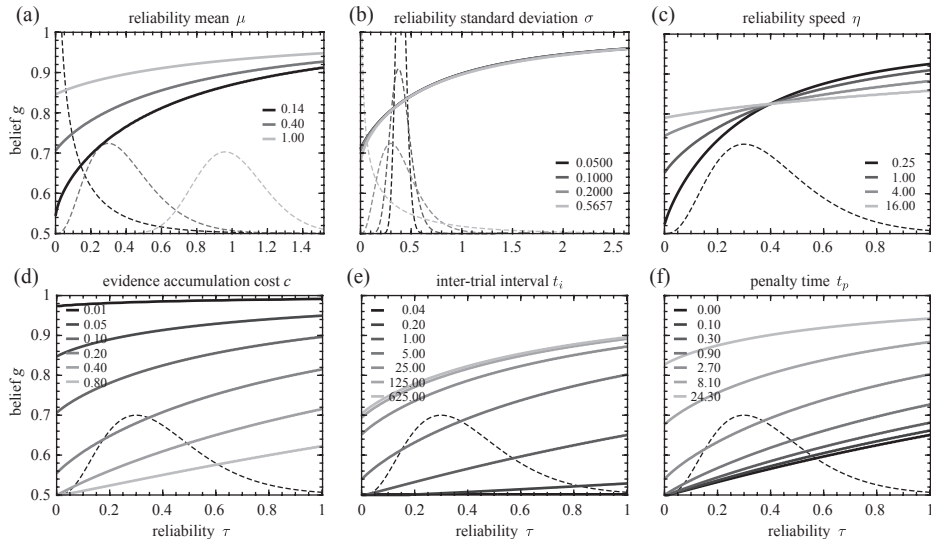

Figure 3: Optimal bounds for different reliability process / task parameters. In the top row, we vary (a) the mean, $\mu$, (b) the standard deviation $\sigma$, or (c) the speed $\eta$ of the CIR process that describes the reliability time-course. In the bottom row, we vary (d) the momentary cost $c$ in an $ER$ task, and, in an $RR$ task (e) the inter-trial interval $t_i$, or (f) the penalty time $t_p$. In all panels, solid lines show optimal bounds, and dashed lines show steady-state densities of $\tau$ (vertically re-scaled).

inter-trial timing also plays an important role. If the inter-trial interval $t_i$ is long, performing well in single trials is more important, as there are fewer opportunities per unit time to gather reward. In fact, for $t_i \to \infty$, the optimal bound in $RR$ tasks becomes equivalent to that of an $ER$ task [3]. For short $t_i$'s, in contrast, quick, uninformed decisions are better, as many of them can be performed in quick succession, and they are bound to be correct in at least half of the trials. This is reflected in optimal bounds that are significantly lower for shorter $t_i$'s (Fig. 3(e)). A larger penalty time, $t_p$, in contrast, causes a rise in the optimal bound (Fig.3(f)), as it is better to make better, slower decisions, if incorrect decisions are penalized by longer waits between consecutive trials.

## 4.3  Performance comparison with alternative heuristics

As previous examples have shown, the optimal policy is — due to its two-dimensional nature — not only hard to compute but might also be hard to implement. For these reasons we investigated if simpler, one-dimensional heuristics were able to achieve comparable performance. We focused on two heuristics in particular. First, we considered standard diffusion models [1, 2] that trigger decisions as soon as the accumulated evidence, $x(t)$ (Eq. (1)), not weighted by $\tau$, reaches one of the time-invariant bounds at $x_\theta$ and $-x_\theta$. These models have been shown to feature optimal performance when the evidence reliability is constant within single trials [2, 3], and electrophysiological recordings have provided support for their implementation in neural substrate [14, 15]. Diffusion models use the unweighted $x(t)$ in Eq. (1) and thus do not perform Bayes-optimal inference if the evidence reliability varies within single trials. For this reason, we considered a second heuristic that performs Bayes-optimal inference by Eq. (2), with time-invariant bounds $X_\theta$ and $-X_\theta$ on $X(t)$. This heuristic deviates from the optimal policy only by not taking into account the bound's dependence on the current reliability, $\tau$.

We compared the performance of the optimal bound with the two heuristics exhaustively by discretizing a subspace of all possible reliability process parameters. The comparison is shown only for the $ER$ task with accumulation cost $c = 0.1$, but we observed qualitatively similar results for other accumulation costs, and $RR$ tasks with various combinations of $c$, $t_i$ and $t_p$. For a fair comparison, we tuned for each set of reliability process parameters the bound of each of the heuristics such that it maximized the associated $ER$ / $RR$. This optimization was performed by the Subplex algorithm [16] in the NLopt tookit [17], where the $ER$ / $RR$ was found by Monte Carlo simulations.

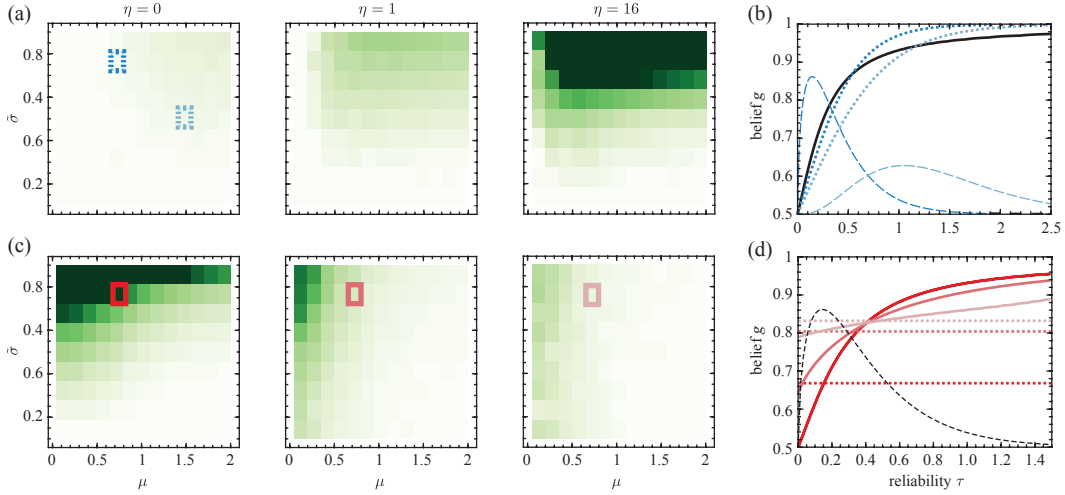

Figure 4: Expected reward comparison between optimal bound and heuristics. (a) shows the reward rate difference (white = no difference, dark green = optimal bound $\geq 2\times$ higher expected reward) between optimal bound and diffusion model for different $\tau$-process parameters. The process SD is shown as fraction of the mean (e.g. $\mu = 1.4, \tilde{\sigma} = 0.8$ implies $\sigma = 1.5 \times 0.8 = 1.12$). (b) The optimal bound (black, for $\eta = 0$ independent of $\mu$ and $\sigma$) and effective tuned diffusion model bounds (blue, dotted curves) for speed $\eta = 0$ and two different mean / SD combinations (blue, dotted rectangles in (a)). The dashed curves show the associated $\tau$ steady-state distributions. (c) same as (a), but comparing optimal bound to constant bound on belief. (d) The optimal bounds (solid curves) and tuned constant bounds (dotted curves) for different $\eta$ and the same $\mu$ / $\sigma$ combination (red rectangles in (c)). The dashed curve shows the steady-state distribution of $\tau$.

### 4.3.1 Comparison to diffusion models

Figure 4(a) shows that for very slow process speeds (e.g. $\eta = 0$), the diffusion model performance is comparable to the optimal bound found by dynamic programming. At higher speeds (e.g. $\eta = 16$), however, diffusion models are no match for the optimal bound anymore. Their performance degrades most strongly when the reliability SD is large, and close to the reliability's mean (dark green area for $\eta = 16$, large $\tilde{\sigma}$, in Fig. 4(a)). This pattern can be explained as follows. In the extreme case of $\eta = 0$, the evidence reliability remains unchanged within single trials. Then, by Eq. (2), we have $X(t) = \tau x(t)$, such that a constant bound $x_\theta$ on $x(t)$ corresponds to a $\tau$-dependent bound $X_\theta = \tau x_\theta$ on $X(t)$. Mapped into belief by Eq. (2), this results in a sigmoidal bound that closely follows the similarly rising optimal bound. Figure 4(b) illustrates that, depending on the steady-state distribution of $\tau$, the tuned diffusion model bound focuses on approximating different regions of the optimal bound.

For a non-stationary evidence reliability, $\eta > 0$, the relation between $X(t)$ and $x(t)$ changes for different trajectories of $\tau(t)$. In this case, the diffusion model bounds cannot be directly related to a bound in $X(t)$ (or, equivalently, in belief $g(t)$). As a result, the effective diffusion model bound in belief fluctuate strongly, causing possibly strong deviations from the optimal bound. This is illustrated in Fig. 4(a) by a significant loss in performance for larger process speeds. This loss is most pronounced for large spreads of $\tau$ (i.e. a large $\sigma$). For small spreads, in contrast, the $\tau(t)$ remains mostly stationary, which is again well approximated by a stationary $\tau$ whose associated optimal policy is well captured by a diffusion model bound. To summarize, diffusion models approximate well the optimal bound as long as the reliability within single trials is close-to stationary. As soon as this reliability starts to fluctuate significantly within single trials (e.g. large $\eta$ and $\sigma$), the performance of diffusion models deteriorates.

### 4.3.2 Comparison to a bound that does not depend on evidence reliability

In contrast to diffusion models, a heuristic, constant bound in belief (i.e. either in $X(t)$ or $g(t)$), as used in [8], causes a drop in performance for slow rather than fast changes of the evidence reliability.

This is illustrated in Fig. 4(c), where the performance loss is largest for $\eta = 0$ and large $\sigma$, and drops with an increase in $\eta$, $\sigma$, and $\mu$.

Figure 4(d) shows why this performance loss is particularly pronounced for slow changes in evidence reliability (i.e. low $\eta$). As can be seen, the optimal bound becomes flatter as a function of $\tau$ when the process speed $\eta$ increases. As previously mentioned, for large $\eta$, this is due to the current reliability providing little information about future reliability. As a consequence, the optimal bound is in these cases well approximated by a constant bound in belief that completely ignores the current reliability. For smaller $\eta$, the optimal bound becomes more strongly dependent on the current reliability $\tau$, such that a constant bound provides a worse approximation, and thus a larger loss in performance.

The dependence of performance loss on the mean $\mu$ and standard deviation $\sigma$ of the steady-state reliability arises similarly. As has been shown in Fig. 3(a), a larger mean reliability $\mu$ causes the optimal bound to become flatter as a function of the current reliability, such that a constant bound approximation performs better for larger $\mu$, as confirmed in Fig. 4(c). The smaller performance loss for smaller spreads of $\tau$ (i.e. smaller $\sigma$) is *not* explained by a change in the optimal bound, which is mostly independent of the exact value of $\sigma$ (Fig. 3(b)). Instead, it arises from the constant bound focusing its approximation to regions of the optimal bound where the steady-state distribution of $\tau$ has high density (dashed curves in Fig. 3(b)). The size of this region shrinks with shrinking $\sigma$, thus improving the approximation of the optimal bound by a constant, and the associated performance of this approximation. Overall, a constant bound in belief features competitive performance compared to the optimal bound if the evidence reliability changes rapidly (large $\eta$), if the task is generally easy (large $\mu$), and if the reliability does not fluctuate strongly within single trials (small $\sigma$). For widely and rapidly changing evidence reliability $\tau$ in difficult tasks, in contrast, a constant bound in belief provides a poor approximation to the optimal bound.

## 5   Discussion

Our work offers the following contributions. First, it pushes the boundaries of the theory of optimal human and animal decision-making by moving towards more realistic tasks in which the reliability changes over time within single trials. Second, it shows how to derive the optimal policy while avoiding the methodological caveats that have plagued previous, related approaches [3]. Third, it demonstrates that optimal behavior is achieved by a bound on the decision maker's belief that depends on the current evidence reliability. Fourth, it explains how the shape of the bound depends on task contingencies and the parameters that determine how the evidence reliability changes with time (in contrast to, e.g., [18], where the utilized heuristic policy is independent of the $\tau$ process). Fifth, it shows that alternative decision-making heuristics can match the optimal bound's performance only for a particular subset of these parameters, outside of which their performance deteriorates.

As derived in Eq. (2), optimal evidence accumulation with time-varying reliability is achieved by weighting the momentary evidence by its current reliability [8]. Previous work has shown that humans and other animals optimally accumulate evidence if its reliability remains constant within a trial [5, 3], or changes with a known time-course [8]. It remains to be clarified if humans and other animals can optimally accumulate evidence if the time-course of its reliability is not known in advance. They have the ability to estimate this reliability on a trial-by-trial basis[19, 20], but how quickly this estimate is formed remains unclear. To this respect, our model predicts that access to the momentary evidence is sufficient to estimate its reliability immediately and with high precision. This property arises from the Wiener process being only an approximation of physical realism. Further work will extend our approach to processes where this reliability is not known with absolute certainty, and that can feature jumps. We do not expect such process modifications to induce qualitative changes to our predictions.

Our theory predicts that, for optimal decision-making, the decision bounds need to be a function of the current evidence reliability, that depends on the parameters that describe the reliability time-course. This prediction can be used to guide the design of experiments that test if humans and other animals are optimal in the increasingly realistic scenarios addressed in this work. While we do not expect our quantitative prediction to be a perfect match to the observed behavior, we expect the decision makers to qualitatively change their decision strategies according to the optimal strategy for different reliability process parameters. Then, having shown in which cases simpler heuristics fail to match the optimal performance allows us focus on such cases to validate our theory.

## Footnotes

[1]We restrict ourselves to $\mu > \sigma$, in which case $\tau(t) > 0$ (excluding $\tau = 0$) is guaranteed for all $t \geq 0$.

[2]The subscript $\cdot_\theta$ indicates the relation to the optimal decision bound $\theta$.

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
