[Supplementary Material]

# Supplement for Optimal decision-making with time-varying reliability

**Jan Drugowitsch**[1]          **Rubén Moreno-Bote**[2]          **Alexandre Pouget**[1]

[1]Dépt. des Neurosciences Fondamentales
Université de Genève
CH-1211 Genève 4, Switzerland
jdrugo@gmail.com,
alexandre.pouget@unige.ch

[2]Research Unit, Parc Sanitari
Sant Joan de Déu and
University of Barcelona
08950 Barcelona, Spain
rmoreno@fsjd.org

## 1   The generative model

Within a single trial, a binary hidden variable $z \in \{-1, 1\}$ (with units $s^{-1}$, if time is measured in seconds) generates a stream of momentary evidence $\mathrm{d}x(t)$, $t \geq 0$, by

$$\mathrm{d}x = z\mathrm{d}t + \frac{1}{\sqrt{\tau(t)}}\mathrm{d}W, \quad \text{where } \mathrm{d}\tau = \eta\left(\mu - \tau\right)\mathrm{d}t + \sigma\sqrt{\frac{2\eta}{\mu}}\sqrt{\tau}\mathrm{d}B, \tag{1}$$

where $\mathrm{d}W$ and $\mathrm{d}B$ are independent Wiener processes. The *reliability* $\tau(t)$ controls how informative the *momentary evidence* $\mathrm{d}x(t)$ is about $z$. $\tau(t)$ follows the Cox-Ingersoll-Ross (CIR) process with mean $\mu$, standard deviation $\sigma$, and speed $\eta$, and has a gamma stead-state distribution with shape $\mu^2/\sigma^2$ and scale $\sigma^2/\mu$ [1].

## 2   Inferring $\tau(t)$ from momentary evidence

It is possible to infer the reliability, $\tau(t)$, instantaneously by making observations of the diffusion process, $x(t)$. To show this, consider the discretization of this diffusion process $\delta x_n = z\delta t + \zeta_n\sqrt{\delta t}\eta_n$, where $\delta t$ is a very small time interval, $\zeta_n^2 = \zeta^2(n\delta t) = \tau(n\delta t)^{-1}$ is a time-dependent variance (inverse of the time-dependent reliability evaluated at $t = n\delta t$), and $\eta_n$ is a zero-mean unit-variance normal random variable independent across time. Now, let us consider the square of the steps $\delta x_n$, which takes the form $\delta x_n^2 = z^2\delta t^2 + \zeta_n^2\delta t\eta_n^2 + 2z\zeta_n\sqrt{\delta t^3}\eta_n$. To estimate the variance $\zeta^2(t)$ we will need to know the following moments of the squared process:

$$\left\langle \delta x_n^2 \right\rangle = \zeta_n^2\delta t + \mathcal{O}(\delta t^2), \tag{2}$$

$$\mathrm{var}\left(\delta x_n^2\right) = 2\zeta_n^4\delta t^2 + \mathcal{O}(\delta t^3), \tag{3}$$

$$\left\langle \delta x_n^2 \delta x_m^2 \right\rangle = \zeta_n^2\zeta_m^2\delta t^2 + \mathcal{O}(\delta t^3), \tag{4}$$

where we have used $\left\langle \eta_n^2 \right\rangle = \left\langle \eta_n^2\eta_m^2 \right\rangle = 1$ and $\left\langle \eta_n^4 \right\rangle = 3$, and averages $\langle . \rangle$ are respect to the process $dW$ in (Eq. 1) (equivalently respect to the $\eta_n$s), and not respect to $dB$.

Let us consider the estimator $y(t) = \sum_{n=1}^{N} \delta x_n^2$, where the time window $t$ has been split into $N$ equal infinitesimal intervals of length $\delta t$. This estimator has moments

$$\langle y(t) \rangle = \delta t \sum_{n=1}^{N} \zeta_n^2 + \mathcal{O}(\delta t) \xrightarrow{\delta t \to 0} \int_0^t \zeta^2(s) \mathrm{d}s, \tag{5}$$

$$\mathrm{var}(y(t)) = \sum_{n=1}^{N} \langle \delta x_n^4 \rangle + \sum_{mn} \langle \delta x_n^2 \delta x_m^2 \rangle - \langle y \rangle^2$$

$$\xrightarrow{\delta t \to 0} \iint_0^t \zeta^2(s_1) \zeta^2(s_2) \mathrm{d}s_1 \mathrm{d}s_2 - \left( \int_0^t \zeta^2(s) \mathrm{d}s \right)^2 = 0, \tag{6}$$

where we have used $t = N \delta t$, and the fact that averages are only with respect to the diffusion process, not respect to trajectories of $\zeta^2(t)$. Since $y(t)$ is a continuous and differentiable deterministic function of the path of $\zeta^2(t)$, the estimator can be used to give infinitely precise estimates of the variance of the underlying process simply by taking the temporal derivative:

$$\frac{d}{dt} y(t) = \zeta^2(t) = \frac{1}{\tau(t)}. \tag{7}$$

It is important that in the definition of $y(t)$ we do not assume that $\tau(t)$ is constant. However, when computing mean and variance of $y(t)$ across $\mathrm{d}W$, we do use the fact that the samples of $\mathrm{d}W$ are i.i.d (which is true by construction, Eq. (1)). In addition, in the derivation of the mean and variance of $y(t)$ we do not use that the samples of $\tau(t)$ are i.i.d, as we do not take the average over the process $\tau(t)$ (equivalently over $\mathrm{d}B$).

Intuitively, $\tau(t)$ is a continuous process (see Eq. (1)), and therefore there is a finite time resolution T below which $\tau(t)$ can be considered approximately constant. Within that time resolution, one can discretize time with infinitesimally small increments $\delta t$ and get as many samples of $\mathrm{d}W$ as desired (i.i.d. by definition, see Eq. (1)). From these samples one can estimate with arbitrarily high precision the reliability $\tau(t)$ of the process, as formally shown above.

There is a single case in which the argument presented above breaks down: consider the limit in which the volatility is infinity ($\eta = 0$ and pre-factor of $\mathrm{d}B$ in Eq. (1) constant). In this case, $\tau(t)$ is not a continuous process, and then $y(t)$ has a discontinuous derivative. Only in this unrealistic case $\tau(t)$ cannot be estimated with infinite precision.

## 3 Inferring the latent $z$

To infer $z$, we again consider the discretization of the particle diffusion process $\delta x_n \sim \mathcal{N}\left(z \delta t, \tau_n^{-1} \delta t\right)$, which is normal with mean $z \delta t$ and variance $\tau_n^{-1} \delta t$. Then, assuming a uniform prior on $z$, that is $p(z) \propto_z 1$, the posterior $z$ is proportional to

$$p(z | \delta x_{0:t}) \propto_z \prod_n \mathcal{N}\left(\delta x_n | z \delta t, \tau_n^{-1} \delta t\right)$$

$$\propto_z e^{-\sum_n \frac{\tau_n (\delta x_n - z \delta t)^2}{2 \delta t}} \tag{8}$$

$$\propto_z e^{-\frac{z^2}{2} \sum_n \delta t \tau_n + z X(t)}$$

where $\delta x_{0:t}$ denotes all momentary evidence until time $t$, and we have defined $X(t) = \sum_n \tau_n \delta x_n$. Adding the appropriate normalization constant, which is the above summed over $z = 1$ and $z = -1$, causes the terms containing $z^2$ to cancel. When taking $\delta t \to 0$, this results in the posterior belief to be given by

$$g(t) \equiv p\left(z = 1 | \mathrm{d}x_{0:t}\right) = \frac{1}{1 + e^{-2X(t)}}, \qquad \text{where } X(t) = \int_0^t \tau(s) \mathrm{d}x(s). \tag{9}$$

This belief is valid even for the case of a bounded accumulation of evidence, as the introduction of such boundaries does not change the sufficient statistics, $X(t)$ [2, 3].

## 4 Finding the expected future return by solving a PDE

The expected future return $\langle V(g + \delta g, \tau + \delta \tau)\rangle_{p(\delta g, \delta \tau | g, \tau)}$ can be found by the solution to a partial differential equation (PDE). To do so, we define $u(g, \tau, t) \equiv V(g, \tau)$ and $u(g, \tau, t + \delta t) \equiv \langle V(g + \delta g, \tau + \delta \tau)\rangle$, and replace this expectation by its second-order Taylor expansion around $(g, \tau)$. Then, we find that, with $\delta t \to 0$, we have

$$\frac{\partial u}{\partial t} = \left( \frac{\langle \mathrm{d}g\rangle}{\mathrm{d}t} \frac{\partial}{\partial g} + \frac{\langle \mathrm{d}\tau\rangle}{\mathrm{d}t} \frac{\partial}{\partial \tau} + \frac{\langle \mathrm{d}g^2\rangle}{2\mathrm{d}t} \frac{\partial^2}{\partial g^2} + \frac{\langle \mathrm{d}\tau^2\rangle}{2\mathrm{d}t} \frac{\partial^2}{\partial \tau^2} + \frac{\langle \mathrm{d}g\mathrm{d}\tau\rangle}{\mathrm{d}t} \frac{\partial^2}{\partial g \partial \tau} \right) u, \qquad (10)$$

with all expectations implicitly conditional on $g$ and $\tau$. The above allows us to find $u(g, \tau, t + \delta t)$ for some known $u(g, \tau)$.

The boundary conditions at $g \in \{0, 1\}$ are $u(g, \tau, t) = V_d(g) = 1$ for all $t$, where $V_d = \max\{g, 1 - g\}$. For $\tau \to \infty$ we have $u(g, \tau, t) = 1$ for all $t$. At $\tau = 0$, all infinitesimal moments except for $\langle \mathrm{d}\tau\rangle = \eta\mu\mathrm{d}t$ are zero, such that we have a deterministic flow towards $\tau > 0$. The main text justifies the use of these boundary conditions.

### 4.1 The infinitesimal moments of $g$ and $\tau$

The infinitesimal moments of $\tau$ are, by the definition of the generative model, Eq. (1), given by

$$\langle \mathrm{d}\tau | g, \tau\rangle = \eta \left( \mu - \tau\right) \mathrm{d}t, \qquad (11)$$

$$\langle \mathrm{d}\tau^2 | g, \tau\rangle = \frac{2\eta\sigma^2}{\mu}\tau\mathrm{d}t, \qquad (12)$$

where we have only retained terms of $\mathcal{O}(\mathrm{d}t)$. The moments of $g$ are found by assuming a small time step $\delta t$ in which $\delta X_n = \tau_n \left( z\delta t + \tau_n^{-1/2}\delta t^{1/2}\eta_n \right)$, where $\eta_n$ is a zero-mean unit-variance normal random variable. To find $\delta g_n$, we approximate the mapping from $X(t)$ to $g(t)$ (Eq. 9)) by a second-order Taylor series expansion around $X_n$ to find

$$\delta g_n = 2(1 - g_n)g_n \left( \tau_n z\delta t + \sqrt{\tau_n \delta t}\eta_n \right) - 2(1 - g_n)g_n(2g_n - 1) \left( \tau_n z\delta t + \sqrt{\tau_n \delta t}\eta_n \right)^2. \quad (13)$$

Taking $\delta t \to 0$ and only retaining terms of $\mathcal{O}(\mathrm{d}t)$ results in the moments

$$\langle \mathrm{d}g | g, \tau\rangle = 0, \qquad (14)$$

$$\langle \mathrm{d}g^2 | g, \tau\rangle = 4(1 - g)^2 g^2 \tau\mathrm{d}t, \qquad (15)$$

$$\langle \mathrm{d}g\mathrm{d}\tau | g, \tau\rangle = 0. \qquad (16)$$

### 4.2 Solving the PDE by the Alternating Direction Implicit method

Having $\langle \mathrm{d}g\mathrm{d}\tau | g, \tau\rangle = 0$ allows us to use the Alternating Direction Implicit (ADI) method to solve the above PDE. To do so, we discretize $u(g, \tau, \cdot)$ on a grid $g_1, \ldots, g_K$ in steps of $\Delta_g$ for $g$, and $\tau_1, \ldots, \tau_J$ in steps of $\Delta_\tau$ for $\tau$. We set $g_1 = 0$ and $g_K = 1$ for the belief, and $\tau_1 = 0$ and $\tau_J$ to twice the 99th percentile of the steady-state distribution of $\tau$. Furthermore, we define $u_{kj}^n \equiv u(g_k, \tau_j, t)$ and $u_{kj}^{n+1} \equiv u(g_k, \tau_j, t + \mathrm{d}t)$. Then, the above PDE, Eq. (10), can be solved by the ADI method [4] in two steps,

$$u_{kj}^{n+\frac{1}{2}} - u_{kj}^n = \frac{\delta t}{2} \left( \frac{\langle \delta g\rangle}{\delta t} \frac{\partial}{\partial g} + \frac{\langle \delta g^2\rangle}{2\delta t} \frac{\partial^2}{\partial g^2} \right) u_{kj}^{n+\frac{1}{2}} + \frac{\delta t}{2} \left( \frac{\langle \delta \tau\rangle}{\delta t} \frac{\partial}{\partial \tau} + \frac{\langle \delta \tau^2\rangle}{2\delta t} \frac{\partial^2}{\partial \tau^2} \right) u_{kj}^n, \quad (17)$$

$$u_{kj}^{n+1} - u_{kj}^{n+\frac{1}{2}} = \frac{\delta t}{2} \left( \frac{\langle \delta g\rangle}{\delta t} \frac{\partial}{\partial g} + \frac{\langle \delta g^2\rangle}{2\delta t} \frac{\partial^2}{\partial g^2} \right) u_{kj}^{n+\frac{1}{2}} + \frac{\delta t}{2} \left( \frac{\langle \delta \tau\rangle}{\delta t} \frac{\partial}{\partial \tau} + \frac{\langle \delta \tau^2\rangle}{2\delta t} \frac{\partial^2}{\partial \tau^2} \right) u_{kj}^{n+1} \quad (18)$$

where $\delta t$ is the time discretization. For all $k$, $j$, and $n$ but the boundary at $\tau = 0$, the derivatives are approximated by the central finite differences

$$\frac{\partial}{\partial g} u_{kj} \approx \frac{u_{k+1,j} - u_{k-1,j}}{2\Delta_g}, \tag{19}$$

$$\frac{\partial^2}{\partial g^2} u_{kj} \approx \frac{u_{k+1,j} - 2u_{kj} + u_{k-1,j}}{\Delta_g^2}, \tag{20}$$

$$\frac{\partial}{\partial \tau} u_{kj} \approx \frac{u_{k,j+1} - u_{k,j-1}}{2\Delta_\tau}, \tag{21}$$

$$\frac{\partial^2}{\partial \tau^2} u_{kj} \approx \frac{u_{k,j+1} - 2u_{kj} + u_{k,j-1}}{\Delta_\tau^2}. \tag{22}$$

At $\tau = 0$, the only required derivative is $\partial u / \partial \tau$, which we approximate by the right finite difference,

$$\frac{\partial}{\partial \tau} u_{k1} \approx \frac{u_{k2} - u_{k1}}{\Delta_\tau}. \tag{23}$$

In the next two subsections, we deal with computing $u^{n+\frac{1}{2}}$ from $u^n$, and then computing $u^{n+1}$ from $u^{n+\frac{1}{2}}$, separately. In both cases, the computation time is of order $\mathcal{O}(KJ)$, such that the expected return can be computed in time linear in the discretization of the $(g, \tau)$ space.

## 4.3 Moving from $n$ to $n + \frac{1}{2}$

Equation (17) can for all $j = 1, \ldots, J$ be written as the linear system

$$\boldsymbol{L}_j^{n+\frac{1}{2}} \boldsymbol{u}_j^{n+\frac{1}{2}} = \boldsymbol{b}_j^n, \tag{24}$$

where $\boldsymbol{u}_j^{n+\frac{1}{2}}$ is a column vector with $K$ elements $\left( u_{1,j}^{n+\frac{1}{2}}, \ldots, u_{K,j}^{n+\frac{1}{2}} \right)$, $\boldsymbol{b}_j^n$ is a vector of the same size, and $\boldsymbol{L}_j^{n+\frac{1}{2}}$ is a tri-diagonal $K \times K$ matrix. Thus, the above system can for each $j$ be solved for $\boldsymbol{u}_j^{n+\frac{1}{2}}$ in $\mathcal{O}(K)$ time, thus leading to an overall computational time complexity $\mathcal{O}(KJ)$.

For $2 \leq j \leq J$, the matrices $\boldsymbol{L}_j^{n+\frac{1}{2}}$ have elements

$$\left( \boldsymbol{L}_j^{n+\frac{1}{2}} \right)_{km} = \begin{cases} 1 + \frac{\delta t}{2\Delta_g^2} \frac{\langle \delta g^2 \rangle}{\delta t} & \text{if } m = k \text{ for } 2 \leq k \leq K - 1, \\ -\frac{\delta t}{4\Delta_g} \frac{\langle \delta g \rangle}{\delta t} - \frac{\delta t}{4\Delta_g^2} \frac{\langle \delta g^2 \rangle}{\delta t} & \text{if } m = k + 1 \text{ for } 2 \leq k \leq K - 1, \\ \frac{\delta t}{4\Delta_g} \frac{\langle \delta g \rangle}{\delta t} - \frac{\delta t}{4\Delta_g^2} \frac{\langle \delta g^2 \rangle}{\delta t} & \text{if } m = k - 1 \text{ for } 2 \leq k \leq K - 1, \\ 1 & \text{if } k = m \text{ for } k \in \{1, K\}, \\ 0 & \text{otherwise.} \end{cases} \tag{25}$$

In the above, the first three lines specify the diagonal, upper diagonal, and lower diagonal, respectively. The fourth lines is responsible for the boundary condition at the $k \in \{1, K\}$ boundary. The associated vectors $\boldsymbol{b}_j^n$ have elements

$$\left( \boldsymbol{b}_j^n \right)_k = \left( 1 - \frac{\delta t}{2\Delta \tau^2} \frac{\langle \delta \tau^2 \rangle}{\delta t} \right) u_{kj}^n +$$

$$\left( \frac{\delta t}{4\Delta_\tau} \frac{\langle \delta \tau \rangle}{\delta t} - \frac{\delta t}{4\Delta_\tau^2} \frac{\langle \delta \tau^2 \rangle}{\delta t} \right) u_{k,j+1}^n + \left( -\frac{\delta t}{4\Delta_\tau} \frac{\langle \delta \tau \rangle}{\delta t} + \frac{\delta t}{4\Delta_\tau^2} \frac{\langle \delta \tau^2 \rangle}{\delta t} \right) u_{k,j-1}^n, \quad (26)$$

for all $2 \leq k \leq K - 1$ and are set to $\left( \boldsymbol{b}_j^n \right)_k = u_{kj}^n$ otherwise. For the $\tau$ boundaries at $j \in \{1, J\}$, $\boldsymbol{L}_j^{n+\frac{1}{2}}$ is set to $\boldsymbol{L}_j^{n+\frac{1}{2}} = \boldsymbol{I}$. At $j = 0$ (corresponding to $\tau = 0$), $\boldsymbol{b}_1^n$ has elements

$$\left( \boldsymbol{b}_1^n \right)_k = \left( 1 - \frac{\delta t}{2\Delta_\tau} \frac{\langle \delta \tau \rangle}{\delta t} \right) u_{k1}^n + \frac{\delta t}{2} \frac{\langle \delta \tau \rangle}{\delta t} u_{k2}^n, \tag{27}$$

for $2 \leq k \leq K - 1$, and is set to $\left( \boldsymbol{b}_1^n \right)_k = u_{k1}^n$ otherwise. At $j = J$, all elements of $\boldsymbol{b}_J^n$ are set to $\left( \boldsymbol{b}_J^n \right)_k = u_{kJ}^n$ to obey the boundary condition at $j = J$.

## 4.4 Moving from $n + \frac{1}{2}$ to $n + 1$

Equation (18) can for all $k = 1, \ldots K$ be written as the linear system

$$\boldsymbol{L}_k^{n+1} \boldsymbol{u}_k^{n+1} = \boldsymbol{b}_k^{n+\frac{1}{2}}, \tag{28}$$

where $\boldsymbol{u}_k^{n+1}$ is a column vector with $J$ elements $\left( u_{k,1}^{n+1}, \ldots, u_{k,J}^{n+1} \right)$, $\boldsymbol{b}_k^{n+\frac{1}{2}}$ is a vector of the same size, and $\boldsymbol{L}_k^{n+1}$ is a tri-diagonal $J \times J$ matrix. Thus, the above system can for each $k$ be solved for $\boldsymbol{u}_k^{n+1}$ in $\mathcal{O}(J)$ time, thus leading to an overall computational time complexity $\mathcal{O}(KJ)$.

For $2 \leq j \leq J - 1$, the matrices $\boldsymbol{L}_k^{n+1}$ have elements

$$\left( \boldsymbol{L}_k^{n+1} \right)_{jn} = \begin{cases} 1 + \frac{\delta t}{2\Delta_\tau^2} \frac{\langle \delta \tau^2 \rangle}{\delta t} & \text{if } n = j \text{ for } 2 \leq j \leq J - 1, \\ -\frac{\delta t}{4\Delta_\tau} \frac{\langle \delta \tau \rangle}{\delta t} - \frac{\delta t}{4\Delta_\tau^2} \frac{\langle \delta \tau^2 \rangle}{\delta t} & \text{if } n = j + 1 \text{ for } 2 \leq j \leq J - 1, \\ \frac{\delta t}{4\Delta_\tau} \frac{\langle \delta \tau \rangle}{\delta t} - \frac{\delta t}{4\Delta_\tau^2} \frac{\langle \delta \tau^2 \rangle}{\delta t} & \text{if } n = j - 1 \text{ for } 2 \leq j \leq J - 1, \\ 1 + \frac{\delta t}{2\Delta_\tau} \frac{\langle \delta \tau \rangle}{\delta t} & \text{if } j = n = 1, \\ -\frac{\delta t}{2\Delta_\tau} \frac{\langle \delta \tau \rangle}{\delta t} & \text{if } j = 1, n = 2, \\ 1 & \text{if } j = n = J, \\ 0 & \text{otherwise.} \end{cases} \tag{29}$$

In the above, the first three lines specify the diagonal, upper diagonal, and lower diagonal, respectively. The fourth and fifth line follow from the boundary condition at $j = 1$. The sixth line follows from the boundary condition at $j = J$. The associated vectors $\boldsymbol{b}_k^{n+\frac{1}{2}}$ have elements

$$\left( \boldsymbol{b}_k^{n+\frac{1}{2}} \right)_j = \left( 1 - \frac{\delta t}{2\Delta_g^2} \frac{\langle \delta g^2 \rangle}{\delta t} \right) u_{kj}^{n+\frac{1}{2}} +$$

$$\left( \frac{\delta t}{4\Delta_g} \frac{\langle \delta g \rangle}{\delta t} + \frac{\delta t}{4\Delta_g^2} \frac{\langle \delta g^2 \rangle}{\delta t} \right) u_{k+1,j}^{n+\frac{1}{2}} + \left( -\frac{\delta t}{4\Delta_g} \frac{\langle \delta g \rangle}{\delta t} + \frac{\delta t}{4\Delta_g^2} \frac{\langle \delta g^2 \rangle}{\delta t} \right) u_{k-1,j}^{n+\frac{1}{2}} \tag{30}$$

for all $2 \leq j \leq J - 1$, and $\left( \boldsymbol{b}_k^{n+\frac{1}{2}} \right)_j = u_{kj}^{n+\frac{1}{2}}$ otherwise. Due to the boundary conditions at $k \in \{1, K\}$ we have $\boldsymbol{L}_k^{n+1} = \boldsymbol{I}$ for both $k$'s, and associated $\left( \boldsymbol{b}_k^{n+\frac{1}{2}} \right)_j = u_{kj}^{n+\frac{1}{2}}$ for all $j$.