[Reviews · NeurIPS 2014]

Submitted by Assigned_Reviewer_1

"Optimal decision-making with time-varying evidence reliability" proposes a model of evidence accumulation (in humans and other animals) in a new, more flexible setting, whereby the evidence reliability can change on a short time scale, i.e. within a single trial. The authors propose a simple, but flexible model of evidence dynamics, derive the corresponding optimal policy and thoroughly discuss how it changes with each of the model's parameters. The meaning and role of every parameter is well explained and the authors provide additional intuitions for their results. The Supplementary Material is clear and compact and seems to provide all the necessary details of the algorithms used. Finally, the authors present evidence for the superiority of their approach over existing policies, which assume that evidence reliability is constant within a trial.

The article is very well written (just a few minor typos, such as "stead-state", check also lines 405 and 431, "out theory"; I guess you could also add that subscript "theta" stands for "all model parameters", l. 99; it might be also slightly confusing that "z" has no unit, isn't there a time unit hanging loose in Eq. (1)? ). It should be well received by NIPS community (and hopefully by experimental collaborators).

Summary: An original article of great clarity, quality and significance.

Submitted by Assigned_Reviewer_5

The paper studies the optimal decision making policy for situations when the evidence during a trial has time varying reliability, the instantaneous variance of which is assumed to be known. The optimal policy is computed by dynamic programming, which is used to characterize the effects of different model parameters on the decision boundaries. Since the optimal policy is computationally expensive, the authors also investigate the ability of two simple heuristics (based on classic decision models ) to approximate the optimal solution, finding several regimes when the performance of such heuristics is expected to be poor.

Quality:
The main derivations seem sound, however I have a few concerns:
I am not convinced by the argument that the estimation of the reliability can be done instantaneously.
In terms of the derivation in the supplementary info: when discretising time such that N->\inf (\delta -> 0, eq.S6)
the estimation assumes that all the samples obtained on this time scale are iid, that it that the variance is constant at that
time resolution. This does not seem quite right; it should at least depend on the volatility of the process (parameter \eta in the CIR process). I am also unsure of the effects of the later discretization of the state space for DP. More generally, it seems to me that the assumptions used in the model don't completely reflect the constraints that an underlying biological system, would operate (hard to reconcile with previous work e.g. Deneve 2012 showed that when estimating a fixed but unknown sensory reliability can cause dynamical effects, such as time-varying bounds, over reliance on early evidence etc.)

Clarity: the paper is well written -- clear and precise.

Novelty: the specific scenario considered here is new, although I am not sure about its relevance for predicting behaviour. Methodologically, once the reliability is assumed known, the estimation of the values by DP, although not trivial, uses well established methods; putting everything together into a reasonably fast solution is quite useful however.

Significance: potentially interesting for the optimal decision community.

Summary: The paper studies the optimal decision making policy for situations when the evidence during a trial has time varying reliability. The main contribution is providing an efficient solution for computing the optimal policy.

Submitted by Assigned_Reviewer_18

Optimal decision-making with time-varying evidence reliability

In this article, the authors formulate an ideal observer model for making a decision based on accumulated evidence over time, where the evidence reliability varies over time. They assume the reliability of the evidence follows a stochastic process (a Cox-Ingersoll-Ross process) and that there is a cost for waiting to act that is linear in time. The resulting computational model is complex, and so they apply Dynamic Programming and Partial Differential Equation solvers to approximate the model given some data. They then explore the behavior of their model for different scenarios (e.g., increasing/decreasing the standard deviation of the reliability) and outline some intuitions for how the parameters affect model behavior. Finally, they compare model behavior to some simple heuristics to outline why those heuristics fail when evidence reliability is actually variable.

The article tackles a difficult problem that is of interest to many within the NIPS community and makes theoretical advances, although they may be incremental advances. It uses mathematical techniques that are not currently prominent within the NIPS community (e.g., applying partial differential equation solvers to stochastic processes), which could have a positive impact on other areas with equivalent mathematical models.

As a cognitive scientist with a moderate background in stochastic process theory, the mathematics in the article are beyond my ability to evaluate properly (beyond cursory reasonableness and consistency). So I will leave the critique of the mathematics to other reviewers. From this perspective, I found the article difficult to follow at times, not because it is poorly written (it is actually fairly well written), but because it is dense and the mathematics are difficult. I would have benefited from some more thorough explanations of the underlying assumptions (why a Cox-Ingersoll-Ross process as opposed to other continuous stochastic processes?) and some more thorough explanations of the figures (which are quite dense).

For future work, it might be useful for the authors to explore using a more discontinuous stochastic process for the evidence reliability. Intuitively, it seems like evidence reliability is likely to stay constant except for when there is a change in the environment (in which case there would be a large jump). This is not meant as a criticism (the authors’ approach seems reasonable and to the best of my knowledge, the first of its kind), but rather as a suggestion for future work.

Also, the article feels incomplete because they setup and develop a nice formalism, but never test it or compare it to previous empirical data. However, there is more than enough theoretical work in the article to justify its worth, and so it reflects my own bias. In fact, presenting the article at NIPS could result in useful collaborations to test the model.

Minor comments:
“stead” -> “steady” (in many places, it says “stead” state rather than “steady” state)
Summary: The article tackles a difficult problem that is of interest to many within the NIPS community and makes theoretical advances, although they may be incremental advances. It uses mathematical techniques that are not currently prominent within the NIPS community (e.g., applying partial differential equation solvers to stochastic processes), which could have a positive impact on other areas with equivalent mathematical models.

Submitted by Assigned_Reviewer_41

In their paper “Optimal decision-making with time-varying evidence reliability” the authors derive optimal bounds for decision-making in a drift-diffusion model where the reliability of the stimulus information fluctuates over time. This fluctuation is modelled by a Cox-Ingersoll-Ross process originally proposed to model the evolution of interest rates. The authors compute optimal bounds for two different cost functions given by the expected reward and the reward rate. The optimization itself is mostly an application of dynamic programming. The results of the paper are of interest to the wide community interested in diffusion to bound models.

The only major comment I have is that there is published work on time-dependent diffusion to bound models by Jan Drugowitsch and collegues (2014) that should be mentioned and the model should be compared to. This is also important to establish the degree of originality of the paper.

See http://elifesciences.org/content/early/2014/06/14/eLife.03005

Typos:

L32 one’s
L405 theory of optimal
L431 our theory
Summary: The authors address an interesting question with established methods. However, there is recent published work that they do not consider.
Author Feedback
Author rebuttal: To all reviewers:
We would like to thank all reviewers for their insightful comments. We will fix all typos in the final version of the manuscript. Regarding our methodological contribution, we used a novel combination of well-established techniques to find the optimal policy/bound – not currently prominent within the NIPS community, as reviewer #2 pointed out. In addition, we provide novel conceptual contributions on the efficiency of decision-makers by characterizing the nature, shape, and parameter-dependency of the optimal policy for time-varying reliability, and under which parameter regimes it is distinguishable from simpler heuristics.

To reviewer #1:
The theta subscript denotes relation to the optimal decision bound (in belief or particle space), rather than all model parameters. The latent variable z indeed has units 1/s, if time is measured in seconds s. We will clarify both in the final version of the manuscript.

To reviewer #2:
The NIPS 8-page format, together with our desire to not gloss over mathematical details we deemed essential for reproducing our results, might have made the submission a rather dense read. We hope to separately publish an extended version that goes more into the details of the issues pointed out by the reviewer.
We used a CIR process rather than some other process to describe the evidence reliability time-course as several features make it particularly suitable for this role: (i) in contrast to a Wiener or Ornstein-Uhlenbeck process, it ensures that the evidence reliability remains positive, and thus meaningful, (ii) it has few parameters with distinct semantics (mean, variance, speed), and (iii) it is sufficiently flexible to describe a wide range of phenomena. While technically not a jump-diffusion process, it can feature jump-like reliability changes (e.g. Fig. 2b). However, even when changing/extending the process assumptions, which is topic of future work, the optimal policy remains a bound that depends on the momentary reliability. Such changes to the process will introduce quantitative changes to the bound shape, but will result in the same qualitative predictions.
With respect to a comparison with empirical data, there is to our knowledge no dataset that thoroughly explores the behavior of decision-maker under different changing-reliability scenarios. However, having identified the parameter regimes under which the optimal policy departs from simple heuristics will allow the design of experiments that perform such a distinction.

To reviewer #3:
Drugowitsch et al. (2014) was published on June 14, after the NIPS submission deadline, such that we could not compare it to our work at that time. It is similar to our work in the sense that they find Eq. 2 to perform optimal evidence accumulation for a known evidence reliability time-course. It differs from our work in that it only considers optimality in the evidence-accumulation sense, but not in the expected reward/reward rate sense. Thus, rather than deriving the optimal policy for a reliability time-course that might change between trials, as is central to our work, they assumed a constant heuristic bound on X and a reliability time-course that is known and constant across trials. As we show, such a constant bound might in some cases deviate strongly from the optimal policy. We will add a brief comparison to Drugowitsch et al. (2014) to the final version of the manuscript.

To reviewer #4:
Regarding the infinite precision in the estimate of reliability, it is important that in the definition of y(t) (above Eq. S5) we do not assume that tau(t) is constant. However, when computing mean and var of y(t) across dW, we do use the fact that the samples of dW are i.i.d (which is true by construction, Eq. 1). Note that in the derivation of Eqs.S 2-7 we do not use that the samples of tau(t) are i.i.d, as we do not take the average over the process tau(t) (equivalently over dB).
Intuitively, tau(t) is a continuous process (see Eq. 1), and therefore there is a finite time resolution T below which tau(t) can be considered approximately constant. Within that time resolution, one can discretize time with infinitesimally small dt and get as many samples of dW as desired (i.i.d. by definition, see Eq. 1). From these samples one can estimate with arbitrarily high precision the reliability tau(t) of the process. The formal proof of this intuition is given in Eqs. S2-7.
There is a single case in which the argument presented above breaks down: consider the limit in which the volatility is infinity (eta=0 and pre-factor of dB in Eq. 1 constant). In this case, tau(t) is not a continuous process, and then y(t) has a discontinuous derivative. Only in this unrealistic case tau(t) cannot be estimated with infinite precision.
In the supplement of the final manuscript, we will add a few sentences and simulations to clarify this counter-intuitive result.
We agree, and already pointed out in the Discussion, that the chosen process in not necessarily physically realistic. Estimating tau(t) will involve some time lag and uncertainty. However, modifying our process to make this necessary, which is planned future work, will not change our results qualitatively, but only quantitatively.
Our approach differs from the one in Deneve (2012), which focuses on estimating the momentary evidence reliability, but does not compute the optimal policy. Instead, it uses a constant-reliability policy as a heuristic substitute. As a result, the decision bound only depends on the momentary reliability estimate (Fig. 1b in Deneve (2012)), but not on the reliability process parameters. This stands in contrast to the optimal policy we have worked out, which is strongly dependent on these parameters (Fig. 3, top row), such as, for example, the speed at which this reliability changes over time. This makes our approach predict behavior that differs from that predicted by Deneve (2012), thus making them empirically distinguishable.